# Integrated Analysis of Single-Molecule Real-Time Sequencing and Next-Generation Sequencing Eveals Insights into Drought Tolerance Mechanism of *Lolium multiflorum*

**DOI:** 10.3390/ijms23147921

**Published:** 2022-07-18

**Authors:** Qiuxu Liu, Fangyan Wang, Yang Shuai, Linkai Huang, Xinquan Zhang

**Affiliations:** College of Grassland Science and Technology, Sichuan Agricultural University, Chengdu 611130, China; sicauliuqiuxu@163.com (Q.L.); wfy_zr@163.com (F.W.); syangerr@163.com (Y.S.); huanglinkai@sicau.edu.cn (L.H.)

**Keywords:** *Lolium multiflorum*, SMRT-Seq, drought stress, PacBio, R2R3-MYB

## Abstract

*Lolium multiflorum* is widely planted in temperate and subtropical regions globally, and it has high economic value owing to its use as forage grass for a wide variety of livestock and poultry. However, drought seriously restricts its yield and quality. At present, owing to the lack of available genomic resources, many types of basic research cannot be conducted, which severely limits the in-depth functional analysis of genes in *L. multiflorum*. Therefore, we used single-molecule real-time (SMRT) and next-generation sequencing (NGS) to sequence the complex transcriptome of *L. multiflorum* under drought. We identified 41,141 DEGs in leaves, 35,559 DEGs in roots, respectively. Moreover, we identified 1243 alternative splicing events under drought. LmPIP5K9 produced two different transcripts with opposite expression patterns, possibly through the phospholipid signaling pathway or the negatively regulated sugar-mediated root growth response to drought stress, respectively. Additionally, 13,079 transcription factors in 90 families were obtained. An in-depth analysis of R2R3-MYB gene family members was performed to preliminarily demonstrate their functions by utilizing subcellular localization and overexpression in yeast. Our data make a significant contribution to the genetics of *L. multiflorum*, offering a current understanding of plant adaptation to drought stress.

## 1. Introduction

Drought is a periodic and growing natural disaster that impacts extensive subject areas [1], including water resources [2,3] and crop yield [4,5,6], as well as a range of environmental systems [7], resulting in serious harm to ecological security and human society [8,9]. Over the past half-century, drought has become more and more serious all over the world [10], which has greatly reduced the productivity of grazing grasslands and artificial mowed grasslands.

*L. multiflorum* is one of the most important forage grasses and is widely grown in temperate and subtropical regions worldwide [11]. It is an excellent annual gramineous species recommended for planting, and its forage yield and quality cannot be replaced by other forage grasses at present [12]. It plays an important role in restoring degraded grasslands and establishing artificial grasslands and has both high ecological value and high economic benefit [12]. In recent years, its phytoremediation [13,14,15], bio-ethanol production [16] and anti-inflammatory medicinal properties [17] have also been reported. *L. multiflorum* is a highly self-incompatible plant, corresponding to the complex structure of its genome [18]. Meanwhile, the lack of available genomic resources hinders its improvement by breeders. Thus, there is an urgent need to construct valuable gene data sets and screen key candidate genes in *L. multiflorum*.

At present, vast quantities of data have been generated through second-generation high-throughput sequencing platforms. However, owing to the short reads generated by these technologies, it is quite difficult to obtain full-length sequences, and such sequencing technologies cannot work well in complex regions [19], which limits the ability of researchers to study gene function throughout the entire genome [20]. SMRT sequencing is a third-generation sequencing technology, which is more and more used in full-length sequencing [21,22]. It makes up for the deficiency from short reads and can generate full-length cDNA sequences (4–8 kb on average) without assembly or a reference genome, which greatly increases the potential for the discovery of genes and deeper studies of cell transcription [23,24]. There are many examples of SMRT sequencing being used to explore key genes or pathways to promote molecular breeding. In *Iris halophila*, metal ion transporters were found to be involved in the response to Pb stress using SMRT sequencing [25]. A key synthase in the benzylisoquinoline alkaloid biosynthesis pathway, the main active substance in *Corydalis yanhusuo*, was identified by SMRT sequencing [26]. SMRT sequencing has become an ideal way to construct transcriptomes and analyze novel genetic material, especially for species with no reference genomes.

We utilized SMRT sequencing data generated with the PacBio Sequel platform, which produces long reads, to reveal full-length transcriptome information in *L. multiflorum* under drought. Such SMRT sequencing data can be complemented by NGS short reads [27]. Then, high-quality full-length transcript and drought-regulated genes in multiple tissues of *L. multiflorum* were obtained. On the basis of transcriptome data, we carried out functional annotation, lncRNA prediction, coding sequence prediction, TF analyses and functional validation of candidate genes by subcellular localization and overexpression in yeast. This is the first time to use SMRT sequencing to generate full-length transcription data from *L. multiflorum* under drought and is a very useful and important resource for further research of this important forage. This research can be utilized to elucidate the mechanisms of drought response of *L. multiflorum* and to create drought-tolerant, water-saving and environmentally friendly forage.

## 2. Results

### 2.1. Assembly of the Sequence Datasets and Functional Annotation

We obtained a total of 60.89 Gb of raw data from PacBio ISO-Seq and generated 13,787,867 subreads with a total size of 30.72 Gb. Furthermore, by self-correcting multiple single-molecule sequencing sequences, we obtained 1,026,983 CCS sequences after filtering. Based on their inclusion of 5′-primer, 3′-primer and poly-A tail sequences, CCS sequences were divided into full-length and non-full-length reads. A total of 879,040 full-length reads were found among the CCS sequences, and a total of 834,868 full-length non-chimera (FLNC) reads were obtained through ICE. Finally, LoRDEC software was used to identify sequencing errors, and the RNA-Seq short reads with high accuracy were combined for further correction. A total of 385,645 error-corrected consensus reads were obtained, with an average length of 2571 bp (Table 1, Figure 1C). Finally, CD-HIT software was used to eliminate redundancy among consensus reads, and 385,645 full-length non-redundant transcripts and 207,995 Unigenes of samples were obtained (Table 1). A total of 12,433 transcripts shared by the two samples were found by cluster analysis of transcripts with redundant sequences between the two samples (Figure 1A).

To further investigate gene function, non-redundant sequences were annotated by using CD-HIT. Overall, the transcripts with annotations corresponding to the Nr database were most common, with a total of 182,952 (Figure 1B). Additionally, 77,258 transcripts were annotated in all databases, and 189,898 transcripts were annotated in at least one database (Appendix A).

To investigate whether drought stress affects AS in *L. multiflorum*, 26,169 non-redundant transcripts (Appendix A) were processed into 10,629 UniTransModels (Appendix A), and 1243 AS events (Appendix A) were identified based on UniTransModels. Before and after drought stress, the occurrence of AS events was essentially uniform in both type and quantity. In descending order of quantity, the types can be ranked as follows: retained intron (RI), alternative 3′ splice sites (A3), alternative 5′ splice sites (A5), skipped exons (SE), alternative first exons (AF), and alternative last exons (AL).

### 2.2. Differentially Expressed Genes in L. multiflorum Leaves and Roots under Drought

To evaluate the reliability of the DEG analysis, Pearson correlation analysis was performed for pairs of samples. The coefficients of determination (R2) among the three biological replicates in each condition were at least greater than 0.97, confirming the high correlation among biological replicates and the stability and reliability of the DEG analysis (Figure 2A). In leaves and roots of *L. multiflorum* under drought stress, 41,141 DEGs (19,155 down- and 21,986 up-regulated genes) (Figure 2C) and 35,559 DEGs (17,402 down- and 18,157 up-regulated genes) (Figure 2D) were found, respectively. All DEGs involved in the response to drought stress were analyzed, and 12 significant profiles were obtained by trend analysis (Figure 2B). Profile 15 represents 3955 genes up-regulated in response to drought stress in both leaves and roots (Appendix A). Profiles 14 and 11 represent 3993 up-regulated genes only in leaves and 3397 up-regulated genes only in roots, respectively (Appendix A). Profile 5 represents 4534 genes down-regulated in response to drought stress in both leaves and roots (Appendix A). These results indicated that there were more DEGs in *L. multiflorum* leaves under in vitro drought treatment and that the response of leaves to stress was broader and more active. At the same time, the trend analysis showed that more genes were down-regulated in both leaves and roots.

A comparison of differences between the two treatment groups in leaves and roots (i.e., DRL versus CKL and DRR versus CKR) revealed that 11,940 DEGs responded to drought treatment in both tissues (Figure 3A, Appendix A). To explore the biological functions of DEGs, GO and KEGG enrichment analyses were performed on all the differentially expressed genes in the two treatment groups (Figure 3B,C, Appendix A). Among molecular function (MF) terms, ‘catalytic activity’ and ‘coenzyme binding’ were significantly enriched in both the DRL versus CKL and DRR versus CKR comparisons, simultaneously. Among cellular component (CC) terms, ‘membrane’, ‘1,3-beta-d-glucan synthase complex’, ‘integral component of membrane’, ‘intrinsic component of membrane’ and ‘membrane part’ were significantly enriched in the DRL versus CKL and DRR versus CKR comparisons, simultaneously. No biological process (BP) terms were enriched in both comparisons.

### 2.3. Transcription Factor Statistics and Identification of R2R3-MYB Family Members

Transcription factors play an important role in regulating gene expression and have been an active research focus for decades. Full-length ryegrass transcriptome data were analyzed using the iTAK database, and the transcripts of 6512 transcription factors in 90 families were identified under CK conditions (Figure 4A, Appendix A). A total of 6567 transcription factors in 88 families were identified under DR conditions (Figure 4B, Appendix A). Under CK and DR conditions, the families represented by the most members were SNF2, SET and FAR1 and FAR1, C3H and SNF2, respectively. Under CK and DR conditions combined, four families ranked in the top ten in terms of the number of family members represented, namely SNF2, FAR1, MYB-related and C3H.

Further identification of R2R3-MYB gene family members among MYB-related genes revealed 29 R2R3MYB genes expressed in *L. multiflorum*, sequentially named *LmMYB1* through *LmMYB29*. The ORF lengths of *LmMYB* members ranged from 720 to 2889 bp, and the predicted protein lengths ranged from 239 to 962 amino acids, with molecular weights ranging from 27.23 to 105.65 kDa, respectively (Table 2).

### 2.4. Genetic Analysis of Members of the R2R3-MYB Gene Family

To investigate the phylogenetic relationships of the R2R3MYB family members of *L. multiflorum* and *Arabidopsis thaliana*, a phylogenetic tree was constructed. Thus, R2R3MYB members from the two species could be divided into 13 subgroups, numbered S1 to S13, according to their phylogenetic relationships (Figure 5). In S1, there is only one gene family member, *LmMYB6*, and it is distantly related to all other members. Meanwhile, S3 and S5 subgroups only have members in *L. multiflorum*. All other subgroups consist of members in both *L. multiflorum* and *A. thaliana*.

### 2.5. The Expression Patterns of R2R3-MYB Family Members under Drought

In order to better understand the role of R2R3-MYB proteins in *L. multiflorum* in the response to stress, the expression patterns of LmMYB genes were measured using qRT-PCR. The samples of *L. multiflorum* under drought stress included nine time points (0 min, 15 min, 30 min, 1 h, 2 h, 3 h, 6 h, 12 h, 24 h). The experimental data are briefly summarized as follows. The expression patterns of the 29 LmMYB genes can be roughly divided into three categories: early response (ER), intermediate response (IR) and late response (LR) (Figure 6). The ER genes (10 members) showed a very fast response to drought stress and were significantly upregulated at 15 and 30 min. The IR genes (11 members) were significantly up-regulated at 1–6 h, and some members were continuously upregulated. The (LR) genes (eight members) were up-regulated to a significant level after 6 h and often continued to be up-regulated under subsequent stress.

### 2.6. LmMYB Transcripts Localized to the Nucleus Enhanced Abiotic Stress Tolerance in Yeast

The subcellular localization of *LmMYB1*, *LmMYB8* and *LmMYB9* in tobacco leaves was further studied, revealing that all three genes were expressed in the nucleus, which was consistent with the function of transcription factors (Figure 7A). To further characterize the response of *LmMYB1*, *LmMYB8* and *LmMYB9* to stress, we cloned them, inserted them into the pYES2 vector, and then heterologously expressed them in the INVScI yeast line. All yeast-harboring empty vectors as well as factors containing *LmMYB1*, *LmMYB8* and *LmMYB9* were able to grow normally on SD-URA (2 g/L galactose) medium (Figure 7B). Only the INVScI strains with *LmMYB1*, *LmMYB8* and *LmMYB9* were able to grow on SD-URA (2 g/L galactose) medium under 3 M sorbitol (Figure 7C) and 1.5 M NaCl treatments (Figure 7D). This result indicated that overexpression of *LmMYB1*, *LmMYB8* and *LmMYB9* could significantly improve the resistance of INVScI yeast strains to osmotic stress compared with the strain harboring the empty *pYES2* vector. This also suggests that *LmMYB1*, *LmMYB8* and *LmMYB9* may be involved in the plant response to osmotic stress.

## 3. Discussion

As one of the most extensive forms of stress, drought has a huge impact on the survival of grasses. *L. multiflorum* is a vanguard grass species in artificial grassland construction, a preferred grass species for mowed grasslands and an important grass species for soil restoration. Understanding its response mechanism to drought stress is of substantial value for agricultural production, animal husbandry and ecological restoration. Currently, there is little available genomic information on *L. multiflorum*, so SMRT sequencing and NGS sequencing were used to construct a novel Unigene database for *L. multiflorum* under drought stress in this study. This extensive and comprehensive unigene database provides strong support for future research on the molecular mechanisms of drought responses in *L. multiflorum*. We identified 207,955 Unigenes and 1243 AS events and further characterized a number of key candidate genes involved in the response of *L. multiflorum* to drought stress, which can be used to advance the breeding of *L. multiflorum* to better adapt to increasingly arid environments.

### 3.1. A More Extensive and Complete Transcriptome Dataset

Compared to other *L. multiflorum* transcriptome studies using the Illumina platform, our results provide a more extensive and complete transcriptome dataset, with several critical strengths. First, our full-length transcriptome can later be used as a reference for annotating and assembling subsequent genomes of *L. multiflorum* and related species. Second, 207,955 accurate and high-quality full-length sequences were obtained, providing particularly valuable data for gene structure and gene function analyses. Third, the full-length transcripts generated in this study can be used to study the response of *L. multiflorum* to environmental changes with greater clarity. In this study, 207,955 Unigenes with an average length of 2571 bp were obtained by PacBio SMRT-Seq. Through the combination of Illumina and PacBio platform data, Unigenes are significantly increased in number and length compared with previous studies (Appendix A) [28,29]. The average predicted lengths of Unigenes using Illumina data were only 871 bp and 575.24 bp, respectively. The increase in number and length greatly enriched the *L. multiflorum* unigene library, making it both more extensive and more complete, thus providing solid data support for subsequent gene function studies.

### 3.2. Alternative Splicing Plays an Important Role in Complex Transcriptional Regulation

Alternative splicing plays an important role in stress responses and could enhance transcript diversity dramatically. The phospholipid signaling pathway is involved in regulating plant growth and aging [30]. Additionally, phosphatidylinositol phosphate 5-kinase (PIP5K) is a key part of the phospholipid signaling pathway [31]. PIP5K proteins play important roles in plants and have many functions. For example, *AtPIP5K1* is involved in responses to water stress and the abscisic acid signaling pathway in A. thaliana [32]. *AtPIP5K4* is involved in regulating stomatal opening in *A. thaliana* [33]. In this study, 1243 AS events were identified, and some notable phenomena were found. *LmPIP5K9* produced two different transcripts, i.e., CDS1 (i2_LQ_DE_c113853/f1p10/2988) and CDS2 (i3_LQ_DE_c41394/f1p1/3368), through alternative splicing of a retained intron (RI), each of which may perform different functions in *L. multiflorum* (Figure 8A). Read counts of the two transcripts indicated that the expression levels of CDS1 and CDS2 were low in both leaf and root samples under normal growth conditions, which was consistent with a recent study [34]. This finding implies that *LmPIP5K9* is not required under normal growth conditions. Additionally, CDS1 is rapidly upregulated, about four-fold, after induction by drought stress, which is also corroborated by other studies [34,35]. However, the expression of CDS2, which itself is low under normal growth conditions, is further down-regulated and almost not expressed at all under drought stress. Thus, two transcripts of the same gene have different expression patterns. Accordingly, *LmPIP5K9* may up-regulate the CDS1-mediated phospholipid signaling pathway to participate in the drought stress response. In contrast, CDS2 interacts with a cytosolic invertase to negatively regulate sugar-mediated root growth, as previously reported [36]. In future work, we will confirm the functions of these two transcripts, initially in *A. thaliana*.

### 3.3. Key Distinctive Candidate Genes Involved in the L. multiflorum Drought Stress Response

Drought stress greatly affected gene expression in *L. multiflorum*, and the affected genes exhibited different expression patterns. We combined differential gene analysis, alternative splicing analysis and gene family analysis to identify some distinctive candidate genes involved in the response to drought stress. Homeobox (HOX) genes have been identified and characterized in many eukaryotes and are involved in regulating various aspects of growth and development [37]. We found that HOX22 (i1_LQ_DE_c57408/f1p2/1087) and HOX24 (i3_HQ_DE_c33357/f3p2/3113) had higher expression levels in DRL samples than other samples (Figure 8B). At the same time, there was an enrichment of DEGs in the Intrinsic/Integral component of Golgi membrane identified by GO analysis, suggesting that HOX22/24 responds to drought stress by participating in the formation of the Golgi apparatus. High-affinity K+ transporters (HAK) are present in all plants with known genomes, but not in animals [38]. These proteins play an important role in K+ uptake and transport [39,40] and are also involved in osmotic regulation [41], stabilizing plants by balancing K+ homeostasis during cell growth and drought stress responses. In the present study, the expression of *LmHAK7* in root tissues after drought stress was remarkably high compared with leaf tissues (Figure 8C), which was consistent with a recent study [42]. Characterizing intense responses to drought stress (i.e., fold change DRR:CKR = 71.5) and ultra-high expression in roots (i.e., fold change DRR:DRL = 9.45) will be important focuses of our future work. In the present study, we also found that *LmMYB1*, *LmMYB8* and *LmMYB9* were able to improve drought tolerance and salt tolerance in yeast (Figure 7), suggesting that they might have the same function in plants. Future experiments will aim to genetically transform *A. thaliana* and *L. multiflorum* for functional verification.

## 4. Materials and Methods

### 4.1. Material Cultivation and Sample Collection

In this study, *L. multiflorum* cultivar ‘Chuannong No. 1′ was used. The experimental subjects for transcriptional sequencing included 300 individual plants during their flowering period, all of which were cultivated at the Ya’an research station of Sichuan Agricultural University. In order to obtain more comprehensive full-length transcriptome sequence information, roots, stems, leaves and flowers were collected from 30 randomly selected individual plants in the control group (CK) and in vitro drought treatment group after 6 h (DR).

Samples for expression pattern analysis were planted in pots (25 × 19 × 6 cm) in a growth chamber with day/night cycles of 16 h/8 h and 22/20 °C. When the number of leaves on plants reached three or four, stress treatments were initiated. The 15% PEG6000 is used to simulate drought stress. The leaves were sampled at 0 h, 15 min, 30 min, 1 h, 2 h, 3 h, 6 h, 12 h and 24 h after drought stress. All tissues were immediately stored in liquid nitrogen after sampling and then stored at −80 °C.

### 4.2. Illumina cDNA Library Construction and Next-Generation Sequencing

After RNA was extracted from all samples, 3 μg of RNA was used for Illumina cDNA library construction. Total RNA was extracted by grinding tissue in TRIzol reagent (Invitrogen, Carlsbad, CA, USA) on dry ice and processed following the protocol provided by the manufacturer. For NGS, a total of 12 Illumina cDNA libraries were constructed by using the Illumina Stranded RNA Library Prep Kit (New York, NY, USA). Six libraries were constructed from root tissue and the other six were from leaf tissue. The cDNA library samples were sequenced by Gene Denovo Biotechnology Co. (Guangzhou, China). The quality control of the NGS raw data was conducted as follows: the sequenced raw reads with the adapter and primer sequences and poly-N were filtered out. The clean data were used to correct the PacBio sequencing data in the next step.

### 4.3. PacBio cDNA Library Construction and Single-Molecule Real-Time (SMRT) Sequencing

After RNA was extracted from all samples, 1 μg of RNA was used for PacBio cDNA library construction. Total RNA was extracted from each sample by using TRIzol reagent (Invitrogen, Carlsbad, CA, USA). We constructed two libraries, one from a mixture of roots, stems, leaves and flowers under drought treatment, and the other from a mixture of roots, stems, leaves and flowers under the control treatment. The SMRT sequencing library was prepared with the Clontech SMARTer PCR cDNA Synthesis Kit and the BluePippin Size Selection System protocol. The mRNA enriched by Oligo (dT) magnetic beads was reverse transcribed into cDNA. Then, double-stranded cDNA was generated with the optimum cycle number. In addition, >4 kb size selection was performed using the BluePippinTM Size-Selection System. Then, large-scale PCR was performed for the subsequent SMRTbell library construction. The SMRTbell library was sequenced by Gene Denovo Biotechnology Co. (Guangzhou, China).

The PacBio Sequel sequencing platform, based on SMRT-Seq technology, was used to generate full-length sequence data. The whole data processing process is carried out based on SMRTlink 5.1 (https://www.pacb.com (accessed on 20 May 2020)). Circular consensus sequence (CCS) data were generated from the subread BAM files. CCS.BAM files that were output were then classified into full-length and non-full-length reads using pbclassify.py. The non-full length and full-length fasta files produced were then fed into the cluster step, which performs isoform-level clustering (ICE), followed by final Arrow polishing. Additional nucleotide errors in consensus reads were corrected using Illumina RNA-Seq data with the software LoRDEC. Any redundancy in corrected consensus reads was removed by CD-HIT to obtain final transcripts for the subsequent analysis (Figure 9).

### 4.4. Functional Annotation of PacBio Isoforms

To annotate the isoforms identified in the sequencing data, isoforms were BLASTed against the NCBI non-redundant protein (Nr) database (http://www.ncbi.nlm.nih.gov (accessed on 20 May 2020)), NCBI nucleotide (Nt) database, the Kyoto Encyclopedia of Genes and Genomes (KEGG) database (http://www.genome.jp/kegg (accessed on 20 May 2020)), the Swiss-Prot protein database (http://www.expasy.ch/sprot (accessed on 20 May 2020)) and the COG/KOG database (http://www.ncbi.nlm.nih.gov/COG (accessed on 20 May 2020)) with the BLASTx program (http://www.ncbi.nlm.nih.gov/BLAST/ (accessed on 20 May 2020)) at an E-value threshold of 10^−5^ to evaluate sequence similarity with genes of other species. GO annotation was analyzed using Blast2GO software [43] with Nr annotation isoform results. Then, functional classification of isoforms was performed using WEGO software [44].

### 4.5. Identification of Alternative Splicing Events and lncRNA Prediction

The non-redundant transcripts were processed with the Coding GENome reconstruction Tool [45]. Each transcript family was further reconstructed into one or more unique transcript model(s) (referred to as UniTransModels). Error-corrected non-redundant transcripts were mapped to UniTransModels. Splicing junctions for transcripts mapped to the same UniTransModels were examined, and transcripts with the same splicing junctions were collapsed. Collapsed transcripts with different splicing junctions were detected as transcription isoforms of UniTransModels. Alternative splicing (AS) events were identified with SUPPA (https://github.com/comprna/SUPPA (accessed on 20 May 2020)) [46].

We used the Coding-Non-Coding-Index (CNCI) [47], Coding Potential Calculator (CPC) [48], Pfam-scan [49] and PLEK [50] software tools to predict lncRNA. CNCI profiles adjoin nucleotide triplets to effectively distinguish protein-coding from non-coding sequences independent of known annotations. CPC mainly assesses the extent and quality of the ORF in a transcript and searches the sequences in known protein sequence databases to clarify the coding and non-coding transcripts. Each transcript was translated in all three possible frames and mapped in the Pfam database using Pfam Scan. Any transcript with a Pfam hit was excluded in the following steps. The PLEK support vector machine classifier uses an optimized K-mer approach to construct the best classifier to assess the coding potential for species that lack high-quality genome sequences and annotations. Default parameters were used for four tools. Transcripts predicted with protein-coding potential by any or all of the three tools above were filtered out, and those without any identified coding potential were our candidate set of lncRNAs.

### 4.6. Quantification of Gene Expression Levels and Differential Expression Analysis

Reference sequences were mapped prior to the quantification of gene expression. The reference sequence mapping used CD-HIT to identify the transcripts that are redundant after generating the corrected consensus sequence. The clean reads from RNA-Seq were mapped to the reference sequence. The process used RSEM software [51], and the parameters for bowtie2 were end-to-end and sensitive mode. Default parameters were used for all other settings.

The read count for each transcript was obtained from the mapping results. Differential expression analysis of the two groups was performed using the DESeq2 R package (1.34.0) (http://bioconductor.org/packages/release/bioc/html/DESeq2.html (accessed on 20 May 2020)). The resulting P-values were adjusted using Benjamini and Hochberg’s approach for controlling the false discovery rate. Genes with an adjusted *p*-value < 0.05 as determined by DESeq were identified as differentially expressed.

GO enrichment analysis of differentially expressed genes (DEGs) was conducted using the GOseq R package (1.46.0) (http://bioconductor.org/packages/release/bioc/html/goseq.html (accessed on 20 May 2020)). GO terms with corrected *p*-values < 0.05 were considered significantly enriched among DEGs. KEGG enrichment analysis was conducted as implemented in KOBAS 3.0 (http://kobas.cbi.pku.edu.cn/kobas3 (accessed on 20 May 2020)) with its default parameters.

### 4.7. Transcription Factor Analysis and Identification of R2R3-MYB Gene Family Members

Transcription factors (TFs) are groups of protein molecules that can bind to specific sequences in the 5′ upstream sequences of genes to ensure that the target genes are expressed at a specific intensity at a specific time and location [52]. We used iTAK software [53] to predict plant TFs. R2R3-MYB genes were identified in the PFAM protein family database using HMMER 3.0 software [54], with the MYB-like DNA-binding domains (Pfam, PF00249) as the search query [55] and an initial threshold value of E ≤ 10–10. Basic information about these genes, including PIs, MWs and subcellular localization, was predicted using the ExPASy website (https://web.expasy.org/protparam/ (accessed on 10/02/2021)) [56].

### 4.8. Phylogenetic Analysis

The inferred protein sequences of the R2R3-MYB from *L. multiflorum* and *A. thaliana* were aligned using ClustalW with its default parameters. MUSCLE [57] and Clustal Omega [58] were also used to verify alignment results. The proteins of *A. thaliana* MYB were downloaded from the Arabidopsis Information Resource (TAIR) (http://www.aabidopsis.org/ (accessed on 10 February 2021)). Based on the alignment of the MYB domain, a phylogenetic tree was constructed with MEGA 7.0 [59] using the neighbor-joining (NJ) method. Bootstrap values (>50%) were estimated using 1000 replicates. Interactive Tree Of Life (iTOL) software was used to optimize the obtained phylogenetic tree [60].

### 4.9. Expression Pattern Analysis of the R2R3-MYB Gene Family

Total RNA was extracted from leaves under drought stress. The cDNA was synthesized using a MonScript™ RTIII Super Mix with dsDNase (Two-Step) Kit from Monad Biotech Co., Ltd. (Suzhou, China). Subsequently, real-time quantitative PCR (qRT-PCR) was performed using SYBR qPCR Master Mix (Vazyme, China). Gene-specific primers (shown in Appendix A) were designed to avoid the conserved region. Both eIF4A and HIS3 were used as reference genes [61].

### 4.10. Subcellular Localization and Heterologous Expression

Subcellular localization prediction of gene family members was conducted using the ExPASy website. To confirm the predicted subcellular localization, we inserted the full-length sequences of *LmMYB1*, *LmMYB8* and *LmMYB9* into the pYBA1132 vector. Then, all of them were transformed into tobacco leaves by Agrobacterium-mediated transformation. The primers used are listed in Appendix A. Empty vector GFP was used as a control. The fluorescence images of GFP fusion proteins were observed by FV10 confocal microscopy.

### 4.11. Heterologous Expression of LmMYB1, LmMYB8 and LmMYB9 in Yeast

*LmMYB1*, *LmMYB8* and *LmMYB9* were amplified from the PacBio cDNA library of *L. multiflorum* constructed using the primers listed in Appendix A. The correct CDS regions were cloned into a pYES2 vector for expression in yeast. Cells of the sorbitol- and NaCl-sensitive yeast strain INVScI were obtained from MiaoLing Plasmid Platform. The plasmids empty pYES2, pYES2-LmMYB1, pYES2-LmMYB8 and pYES2-LmMYB9 were transformed into INVScI yeast by Carrier DNA (Vazyme, China).

Yeast transformants were screened using SD-Ura plates. The whole process lasted for 2 days at 28 °C. To analyze sorbitol and NaCl resistance, the validated single colonies were cultured in liquid SD-Ura medium (2% galactose) at 28 °C and 150 rpm. When the medium OD600 value reaches 2.0, gradient dilution is performed (10^−1^, 10^−2^, 10^−3^). Diluted yeast was spotted onto SD-Ura plates containing 3 M sorbitol or 2 M NaCl in turn, which were photographed and observed by the naked eye.

## 5. Conclusions

Through the analysis of transcriptome changes in leaves and roots of annual ryegrass under drought stress using SMRT and NGS sequencing technologies we identified many genes with potentially related functions and revealed complex transcript responses such as alternative splicing and lncRNA expression. These results contribute to the current understanding of the complexity of transcriptional regulation in plant drought responses. In particular, the R2R3-MYB transcription factor family was identified and analyzed using the high-quality full-length transcriptome. The candidate drought response regulatory factors *LmMYB1*, *LmMYB8* and *LmMYB9* in *L. multiflorum* were preliminarily verified in both tobacco and yeast. These results help clarify the mechanism of plant responses to stress. Future studies will focus on plant responses to stress to improve our understanding of plant adaptation to climate change.

## Figures and Tables

**Figure 1 ijms-23-07921-f001:**
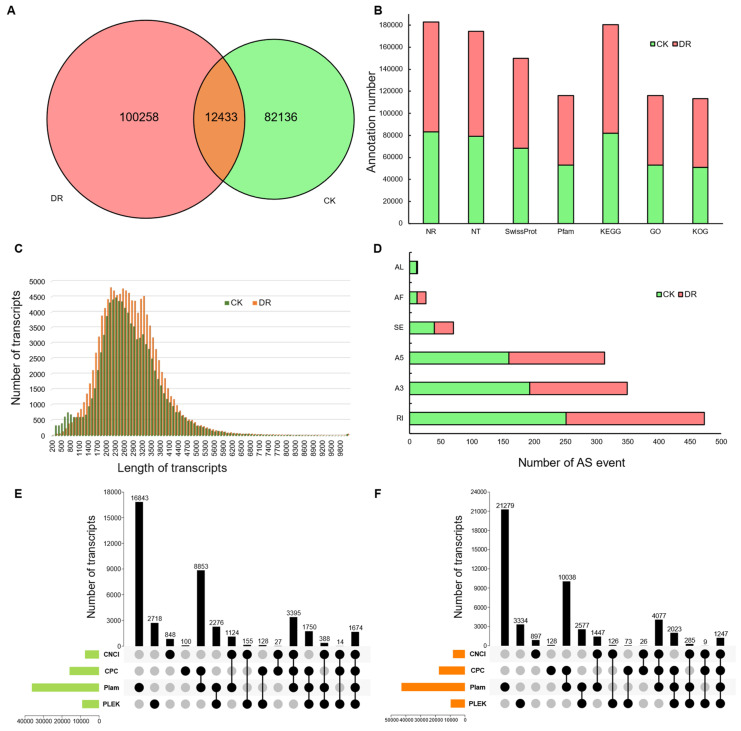
Gene function annotation and gene structure analysis. (**A**) Venn diagram comparing transcripts between control (CK) and drought (DR) conditions. (**B**) Summary of annotation results across the seven databases. (**C**) Length distribution of transcripts. (**D**) Statistical summary of alternative splicing (AS) events in the two samples. (**E**) UpSet graph showing lncRNA predictions for CK conditions. (**F**) UpSet graph showing lncRNA predictions for DR conditions.

**Figure 2 ijms-23-07921-f002:**
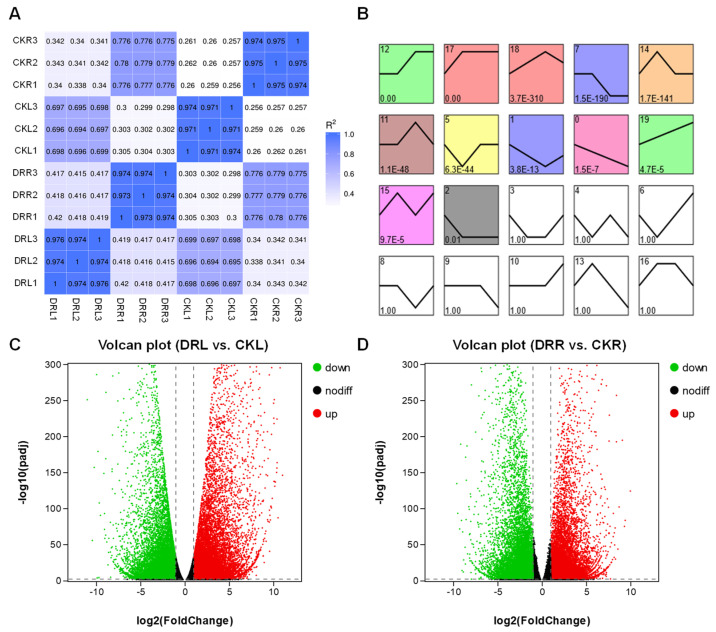
Differentially expressed genes in *L. multiflorum* leaves and roots under drought. (**A**) Pearson correlation coefficients between all samples. CKL and CKR represent leaves and roots under normal control conditions, respectively. DRL and DRR represent leaves and roots under drought stress, respectively. (**B**) Profiles ordered based on the P-value of genes assigned versus expected results. (**C**) Volcano plot of DRL versus CKL. (**D**) Volcano plot of DRL versus CKL. Green represents down-regulated expression, red represents up-regulated expression, and black indicates no significant difference.

**Figure 3 ijms-23-07921-f003:**
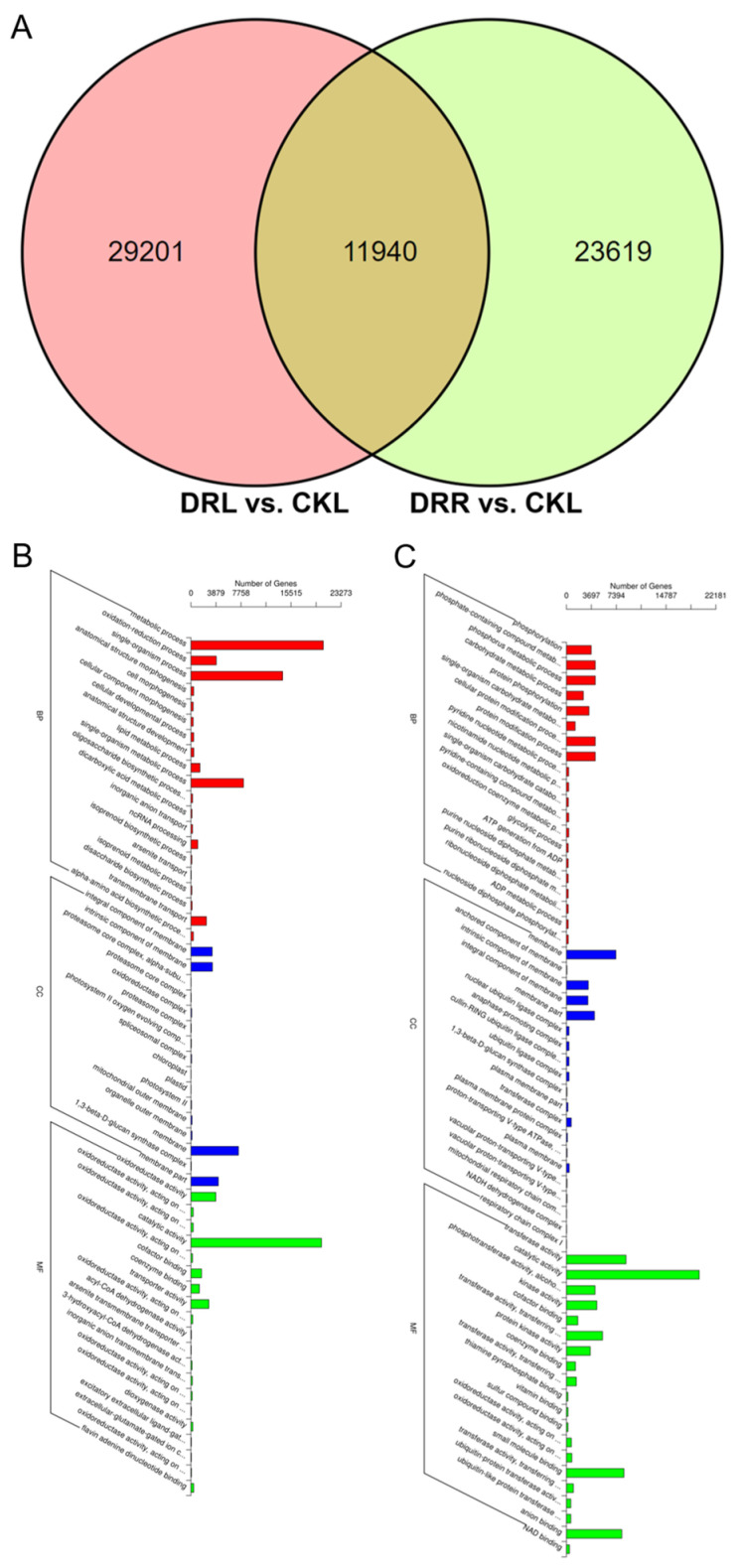
Biological function annotation of differentially expressed genes (DEGs). (**A**) Venn diagram for DEGs in the leaves in control versus drought conditions (DRL versus CKL) and roots in control versus drought conditions (DRR versus CKR) comparisons. (**B**) Gene Ontology (GO) enrichment analysis for DRL versus CKL. (**C**) GO enrichment analysis for DRR versus CKR.

**Figure 4 ijms-23-07921-f004:**
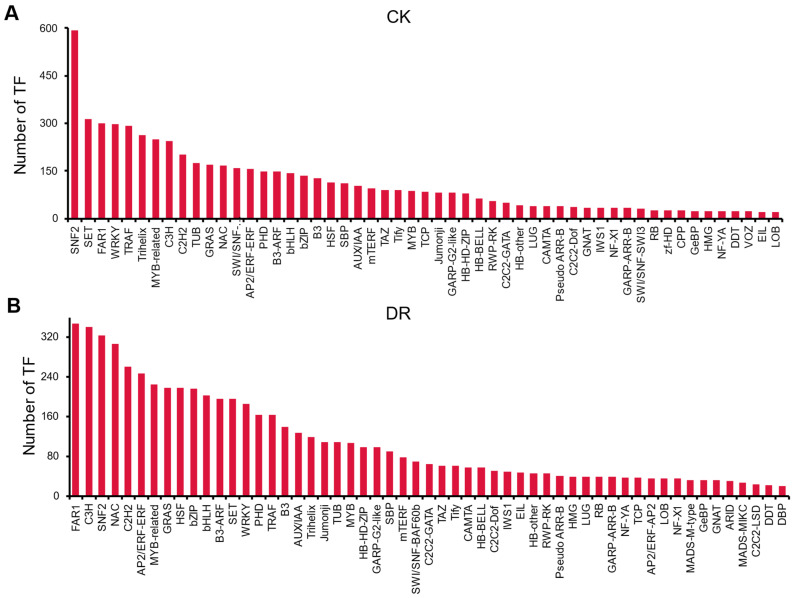
Summary of expressed transcription factor families. (**A**) Transcription factor families in CK. (**B**) Transcription factor families in DR.Transcription factor families are shown along the x-axis, while the corresponding numbers of different transcription factors are represented on the y-axis. Because of the large number of transcription factor families, the histogram shows only those families with more than 20 members.

**Figure 5 ijms-23-07921-f005:**
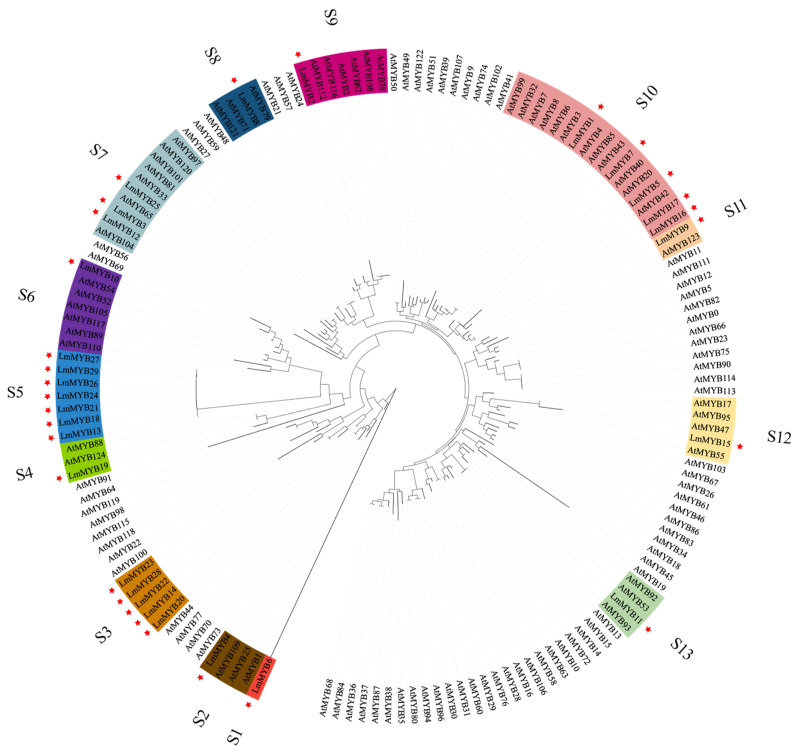
Phylogeny and distribution of R2R3-MYB transcription factors. The phylogenetic tree of R2R3-MYB proteins shows family members from both *Arabidopsis thaliana* and *Lolium multiflorum*. The tree was generated with MEGA 7.0 software using the neighbor-joining (NJ) method based on the inferred amino acid sequences. R2R3-MYB members in *L. multiflorum* are labeled with red stars. S1 to S13 represent each of the different subgroups.

**Figure 6 ijms-23-07921-f006:**
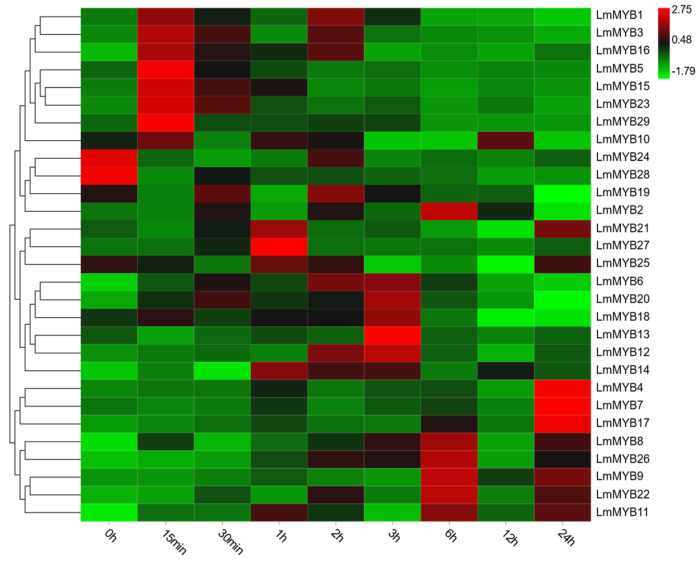
Expression profiles of *LmMYB* genes under drought stress conditions. Red and green represent relatively high and low expression compared to the control, respectively. The values used in the heat map were determined by qRT-PCR and normalized based on the expression of *eIF4A* and *HIS3*. Clustering was conducted by row.

**Figure 7 ijms-23-07921-f007:**
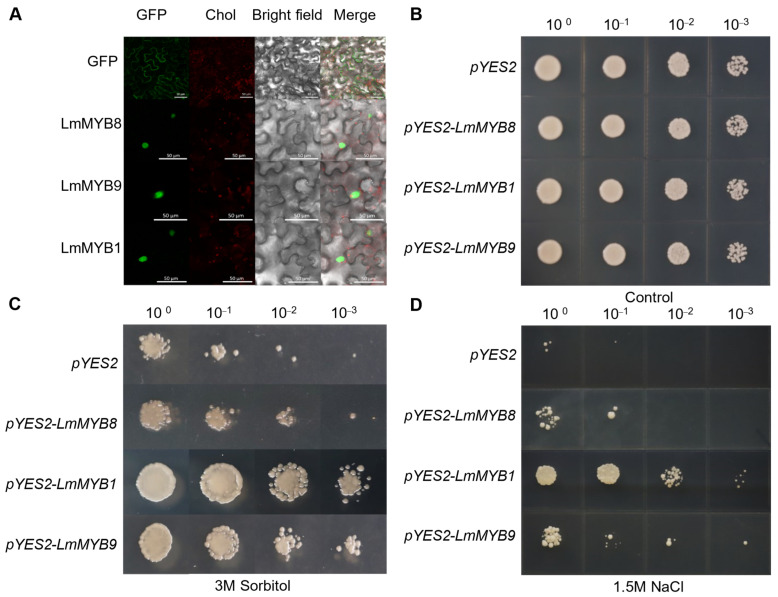
*LmMYB* transcripts are localized to the nucleus and enhanced drought and salt tolerance in yeast. (**A**) Subcellular localization of *LmMYB1*, *LmMYB8* and *LmMYB9*; scale bar, 50 μm. (**B**–**D**) The growth of yeast transformed with the empty vector *pYES2* or with *pYES* harbouring *LmMYB1*, *LmMYB8* or *LmMYB9* under control, 3 M sorbitol and 1.5 M NaCl conditions.

**Figure 8 ijms-23-07921-f008:**
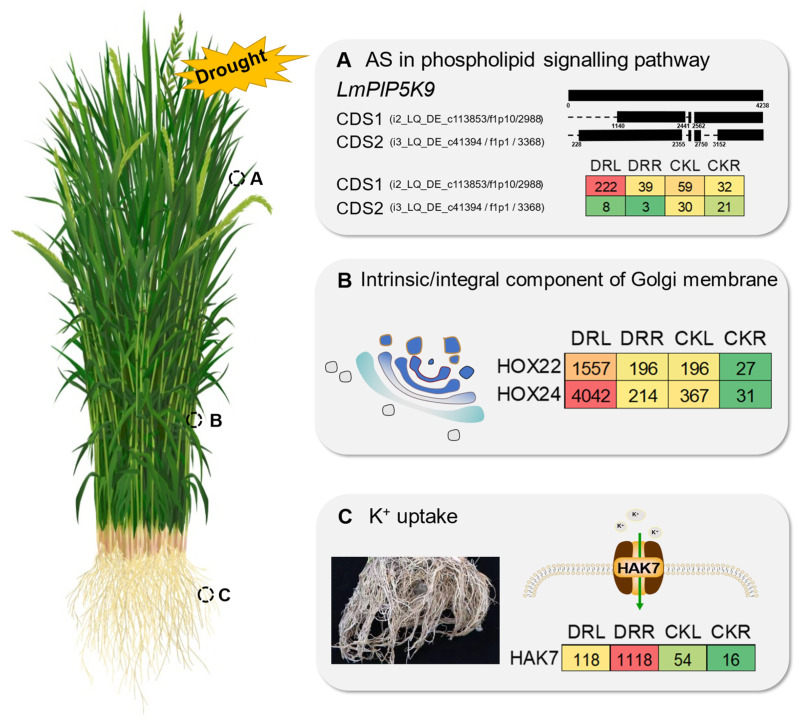
The complexity of transcriptional regulation in the *Lolium multiflorum* drought response. (**A**) The role of alternative splicing (AS) in the phospholipid signaling pathway. The black bars represent sequences, and the grey dotted lines represent gaps. (**B**) The HOX gene family is highly expressed in leaf tissues under drought and participates in intrinsic/integral components of the Golgi membrane. (**C**) The plant-specific gene HAK was highly expressed in root tissue under drought and is involved in K+ uptake. Red and green represent high and low expression, respectively, and the number in each box represents the corresponding read count.

**Figure 9 ijms-23-07921-f009:**
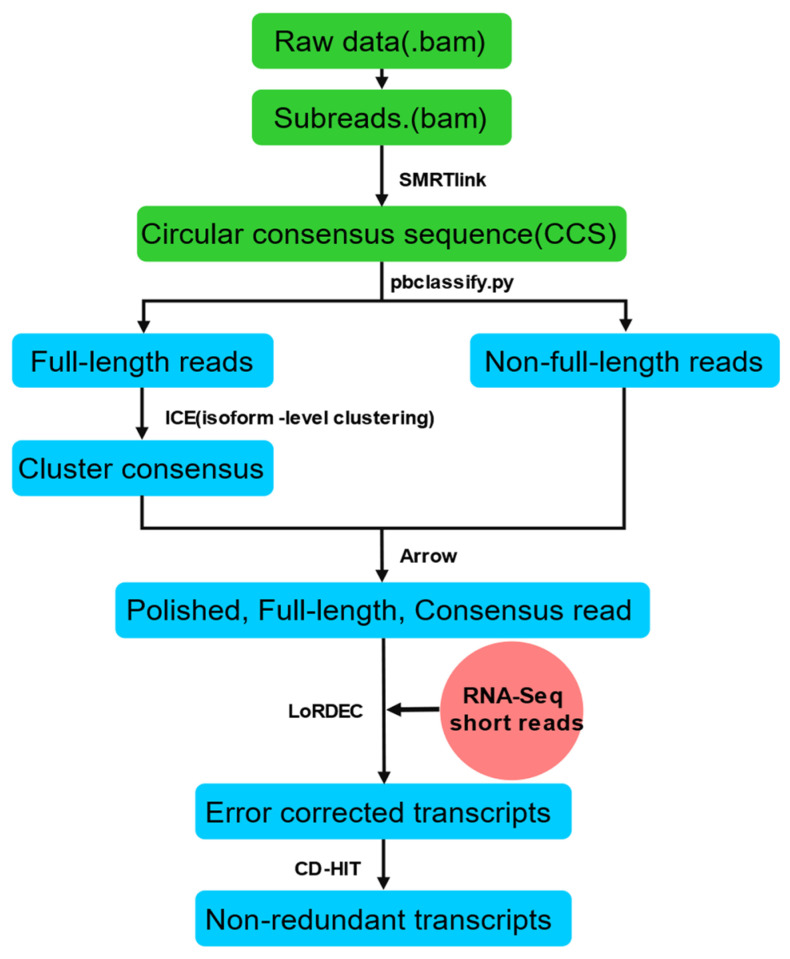
Analysis pipeline for SMRT-Seq data.

**Table 1 ijms-23-07921-t001:** Summary of *Lolium multiflorum* single-molecule real-time sequencing results.

	CK		DR		Total
**Subreads base (G)**	15.48		15.24		30.72
**number**	6,944,546		6,843,321		13,787,867
**Average length (bp)**	2229		2227		2228
**N50 (bp)**	2652		2694		-
**CCS**	488,868		538,115		1,026,983
**5′-primer**	456,294		489,188		945,482
**3′-primer**	462,375		501,744		964,119
**Poly-A**	458,539		497,870		956,409
**Full length**	424,911		454,129		879,040
**FLNC**	403,543		431,325		834,868
	**Before Correction**	**After Correction**	**Before Correction**	**After Correction**	**After Correction**
**Total nucleotides**	469,861,260	471,334,422	519,052,371	520,093,777	991,428,199
**Total number**	184,267	184,267	201,378	201,378	385,645
**Mean length (bp)**	2550	2558	2578	2583	2571
**Min length (bp)**	192	193	200	197	193
**Max length (bp)**	14,449	14,437	14,422	14,348	14,437
**N50 (bp)**	2755	2759	2795	2798	-
**N90 (bp)**	1774	1778	1782	1784	-
**Transcripts Length Interval**	**Number of Transcripts**	**Number of Unigenes**	**Number of Transcripts**	**Number of Unigenes**	
<500 bp	1973	1086	418	211	
500–1k bp	4402	3289	3408	2299	
1–2k bp	46,268	17,602	50,843	25,263	
2–3k bp	80,106	40,389	87,946	45,276	
>3k bp	51,518	32,554	58,763	39,986	
Total	184,267	94,920	201,378	113,035	

**Table 2 ijms-23-07921-t002:** R2R3-MYB family members identified in *Lolium multiflorum*.

	ORF Length (bp)	No. of AA	pI	Mw (kDa)
*LmMYB1*	747	248	9.12	27.75
*LmMYB2*	984	327	5.1	35.71
*LmMYB3*	1155	384	6.68	42.41
*LmMYB4*	1056	351	9.35	37.45
*LmMYB5*	849	282	5.08	31.61
*LmMYB6*	927	308	5.75	31.45
*LmMYB7*	906	301	5.26	33.52
*LmMYB8*	783	260	6.37	29.02
*LmMYB9*	1059	352	5.21	38.58
*LmMYB10*	720	239	7.13	27.23
*LmMYB11*	978	325	5.46	34.48
*LmMYB12*	1371	456	5.06	50.54
*LmMYB13*	1095	364	5.19	40.74
*LmMYB14*	1860	619	6.97	67.38
*LmMYB15*	1287	428	6.19	47.15
*LmMYB16*	873	290	6.39	32.21
*LmMYB17*	756	251	6.09	28.21
*LmMYB18*	915	304	5.27	33.87
*LmMYB19*	2094	697	4.83	75.73
*LmMYB20*	1872	623	6.97	67.75
*LmMYB21*	1089	362	5.18	40.47
*LmMYB22*	2889	962	5.11	105.65
*LmMYB23*	2538	845	5.4	93.78
*LmMYB24*	1002	333	5.21	37.37
*LmMYB25*	1650	549	5.15	59.42
*LmMYB26*	846	281	5.34	31.49
*LmMYB27*	1065	354	5.16	39.49
*LmMYB28*	2556	851	5.48	94.47
*LmMYB29*	933	310	5.11	34.54

Note: ORF, open reading frame; No. of AA, number of amino acids; Mw, molecular weight; pI, isoelectric point.

## Data Availability

The datasets generated in this study have been uploaded to the NCBI database under the accession number PRJNA634598.

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
