# Peer review of "Integrated Analysis of Single-Molecule Real-Time Sequencing and Next-Generation Sequencing Eveals Insights into Drought Tolerance Mechanism of Lolium multiflorum"

_ijms, 2022, doi:10.3390/ijms23147921_

Round 1

Reviewer 1 Report

Dear Authors

The work is very interesting and novel. However, using more genotypes could be better. The English language is good and very clearly understandable. Minor correction with scientific names of species and some question about experimental design below should be addressed. 

Line 55- Iris halophila should be italic. In general, there is some problem with the scientific names that are not italic for example Line 94 or line 109. Please check it in whole manuscript.

Which method/kits were used for RNA extraction?

Authors used 15% of PEG6000 and the latest sampling was 24 h after treatment. I am wondering if 15% and 24h are enough for plants to be under stress? What is the scientific justification showing plants were really under stress?

Line 324: The leaves were sampled at 0 h, 15 min, 30 min, 1 h, 2 h, 3 h, 6 h, 12 h and 24 h after drought stress and in the line 338: We constructed two libraries, one from a mixture of roots, stems, leaves, and flowers under drought treatment. HOW MANY HOURS AFTER DROUGHT TREATMENTS ALL THOSE SAMPLES WERE COLLECTED?

Best wishes

Author Response

Comment 1: Line 55- Iris halophila should be italic. In general, there is some problem with the scientific names that are not italic for example Line 94 or line 109. Please check it in whole manuscript.

Response 1: We are very sorry for our careless mistake and it was rectified at Line 55, 94, 109, 122, 246, 302.

Comment 2: Which method/kits were used for RNA extraction?

Response 2: Thank you for your comments. Total RNA was extracted from each sample by using TRIzol reagent (Invitrogen, CA, USA). At the same time, we also made some modifications at Line 330-332 and Line 341,342.

Comment 3: Authors used 15% of PEG6000 and the latest sampling was 24 h after treatment. I am wondering if 15% and 24h are enough for plants to be under stress? What is the scientific justification showing plants were really under stress?

Response 3: Yes, it is very important point, we observed no significant phenotypic changes within 24h. But plants are already genetically responsive to drought stress with this condition (15%PEG and 24h), as has been reported in ryegrass [1], Triticum durum [2], and tea [3] in recent years. We focused on the expression patterns of transcription factors which regulate gene expression. The results of figure 6 show that members of the R2R3-MYB transcription factor family have varying degrees of expression changes, indicating that plants are responding to drought stress.

Comment 4: Line 324: The leaves were sampled at 0 h, 15 min, 30 min, 1 h, 2 h, 3 h, 6 h, 12 h and 24 h after drought stress and in the line 338: We constructed two libraries, one from a mixture of roots, stems, leaves, and flowers under drought treatment. HOW MANY HOURS AFTER DROUGHT TREATMENTS ALL THOSE SAMPLES WERE COLLECTED?

Response 4: We apologize for not having a clearly described sampling strategy. We have made corresponding modifications at Line 316 to make it clearer. In short, the first paragraph (Line 315-321) describes the sampling strategy of transcriptome sequencing samples. In order to obtain more comprehensive full-length transcriptome sequence information, roots, stems, leaves and flowers were collected from 30 randomly selected individual plants in the control group (CK) and in vitro drought treatment group after 6 h (DR). And the second paragraph (Line 322-327) describes the sampling strategy of expression pattern analysis. The leaves were sampled at 0 h, 15 min, 30 min, 1 h, 2 h, 3 h, 6 h, 12 h and 24 h after drought stress.

[1] Xing J., et al. Genome-Wide Identification and Characterization of the LpSAPK Family Genes in Perennial Ryegrass Highlight LpSAPK9 as an Active Regulator of Drought Stress. Front Plant Sci. 2022 Jun 2; 13:922564.

[2] Faraji, S., et al. The AP2/ERF Gene Family in Triticum durum: Genome-Wide Identification and Expression Analysis under Drought and Salinity Stresses. Genes 2020 Dec 7;11(12):1464.

[3] Zhang Y., et al. CsGSTU8, a Glutathione S-Transferase from Camellia sinensis, Is Regulated by CsWRKY48 and Plays a Positive Role in Drought Tolerance. Front Plant Sci. 2021 Dec 9; 12:795919.

Reviewer 2 Report

The presented manuscript (Integrated analysis of single-molecule real-time sequencing and next-generation sequencing eveals insights into drought tolerance mechanism of Lolium multiflorum) is relevant, very interesting, and has a fundamental value. The obtained results make a significant contribution to the genetics of L. multiflorum, offering a current understanding of plant  drought stress adaptation. 

The results are very informative and presented very clearly. In my opinion, the paper undoubtedly will be of great interest to the readers and may be accepted for publication in the present form.

Author Response

Thank you for your comments.